# Hydrophobized Reversed-Phase Adsorbent for Protection of Dairy Cattle against Lipophilic Toxins from Diet. Efficiensy In Vitro and In Vivo

**DOI:** 10.3390/toxins11050256

**Published:** 2019-05-07

**Authors:** Alexander Sotnichenko, Evgeny Pantsov, Dmitry Shinkarev, Victor Okhanov

**Affiliations:** Research and Production Center “Fox & Co” Ltd.; 117149, Simferopol Boulevard, 8, 117149 Moscow, Russia; pantsov@fox-rpc.com (E.P.); shinkarev@fox-rpc.com (D.S.); company@fox-rpc.com (V.O.)

**Keywords:** mycotoxin, PAH, POP, lipophilicity, Log P, cattle, mastitis, ryegrass staggers, adsorbent, bioaccumulation

## Abstract

The steady growth of inflammatory diseases of the udder in dairy cattle forces us to look for the causes of this phenomenon in the context of growing chemical pollution of the environment and feeds. Within the framework of this concept, an analysis was made of the polarity level of the three toxic impurity groups, which are commonly present in dairy cattle feeds. These impurities are presented by mycotoxins, polyaromatic hydrocarbons (PAH) and persistent organic pollutants (POP). It has been determined that 46% of studied mycotoxins (*n* = 1500) and 100% of studied polyaromatic hydrocarbons (*n* = 45) and persistent organic pollutants (*n* = 55) are lipophilic compounds, prone to bioaccumulation. A comparative evaluation of the sorption capacity of four adsorbents of a different nature and polarity with respect to the simplest PAH, naphthalene and lipophilic estrogenic mycotoxin, zearalenone in vitro has been carried out. The highest efficiency in these experiments was demonstrated by the reversed-phase polyoctylated polysilicate hydrogel (POPSH). The use of POPSH in a herd of lactating cows significantly reduced the transfer of aldrin, dieldrin and heptachlor, typical POPs from the “dirty dozen”, to the milk. The relevance of protecting the main functional systems of animals from the damaging effects of lipophilic toxins from feeds using non-polar adsorbents, and the concept of evaluating the effectiveness of various feed adsorbents for dairy cattle by their influence on the somatic cell count in the collected milk are discussed.

## 1. Introduction

It is known that inflammatory diseases of the udder, which depending on their severity, are usually referred to as subclinical or clinical mastitis, cause serious damage to agriculture [1,2]. The annual economic losses that cause these diseases of dairy cattle on a global scale are account for billions of dollars. To date, the pathogens that provoke the development of mastitis and the factors that determine the sensitivity of animals to this pathology are well determined and studied [2,3,4,5,6,7]. Among them, the properties inherent in the animals themselves, such as the type of animal, genetic features [8], productivity, age, number of calving/lactation, structure of the udder [9,10] etc., as well as external factors associated with the conditions of care, milking and feeding animals [1,2,3,9,10,11,12,13] are usually examined. 

Today we can conclude that despite the considerable progress achieved in selective genetics and technology of the maintenance, care and milking of the animals, the number of cases of registered mastitis continuously increases. Moreover, in some regions there is a significant increase in the incidence of mastitis, especially in its subclinical form (up to 60% of the herd) [2,11]. Perhaps this might be related to the constant increase in environmental pollution that cannot but affect the quality of feed used in dairy farming. 

Supposedly, despite the undisputed achievements in genetic selection and the modern use of the most advanced technologies in dairy production, the greater, if not essential, effect on dairy cattles’ health and productivity, as well as the consumer quality of the dairy products is determined by the quality of the feed used in the dairy farming. This practice demonstrates that the feed rations of poultry, fish and pigs include feeds based on raw grain materials with different supplements, and dairy cattle feed is based on herbaceous plants with a small supplementation of concentrates. Therefore, the feeds of poultry, fish and pigs majorly contain toxins typical to the raw grain materials, mostly mycotoxins, whereas the feeds of dairy cattle contain toxins typical to the mass of green grass. The herbal mass, in addition to mycotoxins, what will be discussed in more detail later, always contains polyaromatic hydrocarbons (PAHs) and persistent organic pollutants (POPs) as impurities. Consequently, it is common to believe that dairy cattle consume more toxic feed than poultry and the pigs. Since the base of the feed ration of dairy cows during the summer period consists of green fodder, and during the winter periods contains silages, concentrates, haylage and straw, then with time the toxins contained in these feeds may negatively impact the general health of the animals, as well as the ability of the nervous, immune, endocrine and digestive systems, the milk somatic cells count (SCC), bacterial contamination of milk and the level of milk production. 

In this paper, we will use terms that describe the properties of chemicals in relation to their environment. It is necessary to take into account that mycotoxins can be attributed either to polar substances that have good solubility in water, therefore they are called polar or hydrophilic, or to non-polar substances that do not dissolve in water and preferably concentrate in low dielectric media. Such substances are usually designated by the terms non-polar, lipophilic, or hydrophobic. However, since among non-polar substances there are compounds with polar constituents that ensure the presence of a dipole moment in the molecule, the terms lipophilic or hydrophobic will be used more often to refer to substances with Log Pow > 3.

The major toxic components of farm animal feeds are considered to be mycotoxins—secondary metabolites of toxicogenic microscopic fungi [14,15,16,17,18,19]. The mycotoxins represent a very wide range of chemical compounds with different composition, structure and biological properties, which are combined only by one factor, their source of origin. 

With the development of analytical techniques and the equipment of specialized and research laboratories, the number of contaminated feed types, as well as the types and number of mycotoxins determined in feeds, continues to grow [20,21,22,23,24,25,26,27,28]. In other words, the wider the spectrum of the mycotoxins analyzed and found in the feeds, the higher portion of the feeds become contaminated with these compounds. At present, more and more specialists incline that almost all feeds contain mycotoxins. It is a question of their nomenclature and concentration. Based on the figures cited in the research on the evaluation of the contamination of the samples of grass and silages from different countries with various types of microscopic mycotoxin producing fungi belonging to more than 20 genera [20,23,26,29], it is reasonable to anticipate that these samples may contain in different quantities up to 500 or more different mycotoxins, and it is almost impossible to determine all of them and their levels in full.

The danger of the presence of mycotoxins in feed for cattle and other farm animals, in addition to the harm posed to livestock health and reduced productivity [30,31], is complemented by the risk of their transfer to animal products—mainly milk [32,33,34,35], eggs and meat [36,37,38,39]—and thus their inclusion in the human food chains [40,41,42]. First of all, this refers to non-polar toxins, capable of bioaccumulation [43].

In addition to mycotoxins, as they are known, cattle feed always contains two more groups of toxic compounds, which as a result of natural phenomena and human activity, are widely distributed in the environment of any region. These include PAHs, for example, naphthalene, benzopyrene, chrysene, benzanthracene, etc., and POPs, such as aldrin, dieldrin, heptachlor, DDT, hexachlorocyclohexane, polychlorinated biphenyls, dioxins, furans, etc. [37,44,45].

It has been established that PAHs and POPs, like lipophilic mycotoxins, are also capable of bioaccumulation [46] and transfer into milk [37,47,48,49,50] in amounts up to 80% of the received dose from feed, especially POPs. In addition, PAHs together with the mycotoxins, but unlike POPs, are attributed to evolutionarily typical substrates for the vertebrate cytochrome P-450 xenobiotic metabolism system and by their presence in feed, they put an additional burden on the liver detoxification system, reducing its ability to neutralize other toxic xenobiotics, first of all—mycotoxins. 

There is a point of view that not only mycotoxins can synergistically enhance the effect of each other [51,52], but also synergistically interact with PAHs and POPs. It was shown that some POPs, for example, 2,3,7,8-tetrachlorodibenzo-p-dioxin, or simply dioxin, can increase the toxic impact of T-2 toxin on rabbits by several times [53]. In this regard, it is rather difficult to estimate the level of synergistic interactions of toxins and their consequences for the organism while there are several tens or hundreds of different mycotoxins, PAHs and POPs in feeds even in low concentrations. Therefore, in modern animal husbandry, it is imperative to use effective means of protecting animals from the harmful effects of toxins in feed.

To reduce the toxic load of feed, in addition to mechanical, chemical and physical methods of controlling mold fungi, adsorbents are often used. They are used in the form of feed additives to remove the mycotoxins from the gastrointestinal tract of animals. The properties and effectiveness of their use in animal husbandry are described in numerous reviews and original articles cited there [54,55,56,57,58,59,60,61]. Most of the feed adsorbents used show a fairly high efficiency in pig and poultry farming in the presence of mostly polar mycotoxins in feed. But, as practice shows, their effectiveness against lipophilic toxins is much lower. Therefore, the development and use of adsorbents, which can also remove lipophilic toxins from the body of an animal, such as hydrophobic mycotoxins, PAHs and POPs, in dairy farming is very urgent.

In this article, after analyzing the properties of three main types of contaminant for cattle feeds—mycotoxins, PAHs and POPs—the effectiveness of a new adsorbent based on a hydrophobized reversed-phase polyoctylated polysilicate hydrogel (POPSH) in binding lipophilic toxins in vitro and in reducing some of the POP’s transfer into milk in vivo was demonstrated. We will also cite recommendations to fight against the non-polar toxins.

## 2. Results

### 2.1. On the Polarity of Mycotoxins

Figure 1 shows data on the partition coefficients of 1500 different mycotoxins and other extralites of fungi and some bacteria which are widely distributed in soils and can be contained in cattle feeds, presented in the form of a histogram. The abscissa axis on the graph shows the values of partition coefficients with a step of 0.5, and the ordinate reflects the number of substances having the value of this parameter in the specified range. Such a distribution can be called the “lipophilicity profile” for a given set of mycotoxins. Data on the calculated values of LogPow (XLog P3-AA [62]) for these compounds in a table form are presented in the Appendix A. The elements of the table (mycotoxins) are arranged in accordance with the increase in the values of their partition coefficients (Log Pow) with a step of 0.1. Alphabetic sorting in the names of substances is followed within the same value of this parameter.

Based on the data in Appendix A, it follows that 90% of all mycotoxins presented have partition coefficients ranging from −0.2 to +7.2. The whole set (*n* = 1500) in accordance with the partition coefficients of its components is characterized by the following parameters: Mean = 2.91, Median = 2.70, Min = −10.0, Max = +10.1. It is also clear that from the entire array containing 1,500 compounds, the share of polar substances (Log Pow < 1) is 213 substances, or 14.2%, the share of moderately lipophilic toxins (1 ≤ Log Pow < 3) is 592 substances, or 39.5%, and lipophilic compounds (Log Pow ≥ 3) are presented by 695 substances, which constitutes 46.3%.

It should be noted that in the process of forming this database and increasing randomly the number of mycotoxins from 200 to 1500, the relative content of non-polar toxins increased from 25% to 45% with an increase in the total number of mycotoxins to 400, and then remained almost constant at the level reached. Thus, it can be stated that more than 45% of mycotoxins studied in this paper are non-polar substances. These data deserve special attention because, as it is known, non-polar substances pose an additional threat to animals because of their capacity for bioaccumulation [43].

### 2.2. On the Polarity of Polyaromatic Hydrocarbons and Persistent Organic Pollutants

It is known that almost all feeds for dairy cattle, especially those based on grassy plants, contain two more types of impurities - PAHs and POPs [37,45,47,48,49,50].

An analysis of the PubChem data [63] showed that, unlike the mycotoxins, all 100% of PAHs (*n* = 45) and 100% of POPs (*n* = 55) belong to lipophilic substances (see Appendix A, respectively). Thus, the most polar of PAHs is naphthalene, characterized by Log Pow = 3.3, and the most polar of POPs, endrin, has a partition coefficient equal to 3.7. Other representatives of these groups of compounds have even higher values of the partition coefficient in the range from 3.7 to 10.0. This means that all of them, like non-polar mycotoxins, are capable of bioaccumulation (bioconcentration), especially POPs [46].

### 2.3. Comparative Sorption Capacity of Adsorbents in Relation to Lipophilic Sorbates

Figure 2 shows the data on the sorption capacity of four different adsorbents with respect to a non-polar simplest PAH, naphthalene (Log Pow = 3.3) and lipophilic infamous estrogenic mycotoxin, zearalenone (Log Pow = 3.6), a) and b), respectively. Everyone knows what damage this mycotoxin causes to the reproduction of cattle and other farm animals. 

Adsorbents of different nature and polarity were used. Adsorbent No. 1 was chosen among aluminosilicate adsorbents, No. 2 consisted of yeast cell walls, No. 3 is an activated carbon, and adsorbent No. 4 is a reversed-phase adsorbent based on a polyoctylated polysilicate hydrogel (POPSH) [24]. The sorption capacity of the most effective adsorbent was taken as 100%.

It can be seen that non-polar adsorbents (No. 3 and 4) are significantly superior to polar ones (No. 1 and 2) in their ability to bind lipophilic sorbates. It can also be noted that adsorbent No. 4 in terms of sorption capacity exceeds activated carbon. For this reason, its capacity in these experiments was taken as 100%.

### 2.4. Effects of POPSH on the Transfer of Chlorinated Pesticides into Milk

As already mentioned, POPs are most prone to transfer to milk. This ability depends on the composition and properties of POPs, but as established, it is significantly higher than that of mycotoxins and PAHs. The reasons for this phenomenon will be discussed later. Table 1 provides data on the effect of adsorbent No. 4 on the concentration of certain chlorinated pesticides from the “dirty dozen”, namely aldrin, dieldrin and heptachlor, in the collected milk from one of the dairy farms of the north-eastern part of the Kaluga region of the Russian Federation.

It can be seen that the use of POPSH in experimental group consisting of 65 animals for 40 days can significantly reduce the content of the specified chlorinated pesticides in raw milk.

## 3. Discussion

### 3.1. On the Polarity of Mycotoxins in Feed

There are many types of classification for mycotoxins, based on their unique features, but the most important for our consideration are the two most general types of classification. The first relies on the ecological niches of fungal producers. Mycotoxins in feed for cattle, depending on the ecological niche of the source of origin, are conventionally divided into “field” ones, which are formed by phytopathogenic fungi during the period of growth and the ripening of herbaceous plants, grain and other forage crops [64,65,66,67]; “pasture” or “grazing” toxins that are produced by endophytic symbiotic fungi during the period of active vegetation and fruiting of some pasture plants in the warm season [23,68,69,70,71,72], and “storage” toxins, which are formed during the storage of plant products infected with saprophytic molds in warehouses in inappropriate storage conditions [26,54,65,67,73]. It should be noted that such a division is not absolutely strict, since some species of fungi from different genera may exhibit properties that are not characteristic of their own kind—“field”, “pasture” or “storage”.

The second general type of the classification of mycotoxins is founded on their physical and chemical properties [24]. The main criteria of selection in this approach is one of the fundamental characteristic, pertinent to all, without any exclusions, organic compounds, the degree of polarity of a compound, which can be quantitatively described using the partition coefficient of a chemical in the octanol/water system, which is expressed as a decimal logarithm (Log Pow). “Pow” refers to the “partition octanol/water” [74,75]. Further, taking into account the fact that in tables of this article the calculated values for the partition coefficients are given (XLogP3-AA) [62], for simplicity in the text we will use the notation Log Pow. The quantitative assessment inherent in this method of classification has certain advantages over other types of mycotoxin classification methods. Within the confines of such approach, and in accordance with widely accepted schema, all mycotoxins could be conditionally, but univocally, divided into three groups, based on the degree of their polarity, such as: 1) polar (hydrophilic) (Log Pow < 1); 2) moderately hydrophilic/lipophilic (1 ≤ Log Pow < 3); and 3) lipophilic (hydrophobic) (Log Pow ≥ 3). Therefore, any mycotoxin from “field”, “pasture” or “storage” type, regardless of the ecological niche of the producer and biological properties of the substance, can occur in any of the polarity groups depending only on its chemical composition and the structure of the molecule. This approach to the systematization of mycotoxins is not associated with other methods of classifying mycotoxins but may be the most useful in the development of adsorbents to combat these toxins.

As mentioned in the results of the study, the proportion of lipophilic mycotoxins from a sample of 1500 units exceeds 45%. A similar distribution of mycotoxins by their degree of polarity is also observed in real feeds for cattle. Graphically, this can be illustrated by analyzing the data presented in a large-scale study of the prevalence of 139 individual mycotoxins and other fungal and some bacterial metabolites in 86 samples of various cattle feeds and their components obtained from different countries [26]. 

Mycotoxins, which were quantified in this work, significantly differed in the degree of lipophilicity in the range of partition coefficients from −2.3 (deoxynivalenol-3-glucoside) to +8.6 (calphostin C). It was also noted that the majority of feed samples contained from 25 to 40 different mycotoxins. The “lypophylicity profile” in Figure 3 presents data on 22 different mycotoxins, by which more than 60% of all the feed samples studied were contaminated. The X axis shows individual metabolites located on the scale according to the degree of their lipophilicity, or Log Pow value, and along the Y axis, the number of feed samples contaminated by this mycotoxin, as a percentage of the total number of samples examined, are presented. Calculations show that four mycotoxins (deoxynivalen-3-glucoside, nivalenol, deoxynivalen and monoliformin) are 18.2% and can be attributed to polar toxins, seven more mycotoxins (brevianamide F, ergomethrine, tryptophol, tentoxin, tenuazonic acid, emodin and alternariol) constitute 31.8% and can be attributed to moderately lipophilic toxins. The last eleven mycotoxins (alternariol methyl ether, culmorin, aurofuzarin, zearalenone, apicidin, equisetin, enniatins A, A1, B, and B1 and beauvericin) constitute the remaining 50% and belong to lipophilic toxins. At the same time, the degree of contamination of feed with the most hydrophobic mycotoxins, eniathins and beauvericin (Log Pow = 6.5–8.4), was found to be from 87% to 98% [26]. 

The danger of these mycotoxins presenting in cattle feed is that due to their high lipophilicity they are capable of bioaccumulation, and despite the relatively low acute toxicity to vertebrates, they have strong antibiotic properties against a wide range of microorganisms and can modify functional microflora of the rumen, thus violating the digestion of ruminants [76,77,78,79]. In addition to their antibiotic properties, enniatins and beauvericins have a cytotoxic effect on mammalian cells [80,81] and can inhibit the immune system [82,83,84].

An even higher content of lipophilic mycotoxins in feed can be noted when analyzing the degree of contamination of pasture grass samples (*n* = 106) in temperate grasslands of Chako province in Argentina and quantifying 77 mycotoxins in them [20]. Among all mycotoxins detected during the study, only 21 of them showed degree of contamination exceeding 60% of the total number of samples (*n* = 21). Eight of these, or 38%, were moderately lipophilic toxins, and thirteen of them, or 62% were presented by lipophilic mycotoxins. The contamination of grass samples with polar toxins was significantly lower than 60%. Among moderately lipophilic toxins, contamination of samples with emodin, alternariol and monocerin was up to 100%, while contamination with such lipophilic toxins as aurofuzarin, sterigmatocystin, chrysophanol, equisetin, skirin and beauvericin ranged from 90% to 100%. 

It should also be noted that well-known “pasture” endophytic mycotoxins—lolitremes (*n* = 11 in Appendix A), which cause the so-called “ryegrass staggers” [70] also belong to non-polar toxins (Log Pow = 3.9–6.0), as well as the majority of other tremorgenic mycotoxins (see Appendix A).

For this reason, we believe that it is lipophilic toxins that pose the greatest danger to the livestock of dairy cows. Hydrophilic mycotoxins, like other polar toxins, usually dissolve well in water and can be removed from the body through urine, while lipophilic toxins cannot be excreted in the urine, but are usually accumulated in adipose tissue.

At present, work is underway to bring into better compliance mycotoxin producing fungi with their metabolites, presented in Appendix A. Upon its completion, data will be presented on the polarity of mycotoxins and other metabolites, which are produced by micromycetes of a particular genus affiliation. We hope that after completion of this study, it will be possible, at least within Appendix A, to determine which fungi in plants are the most dangerous from the point of view of contamination of feed with non-polar mycotoxins. At this stage of the study, it is already becoming obvious that the proportion of non-polar metabolites produced by “pasture” endophytic fungi from *Acremonium, Aureobasidium, Chaethomium, Cladosporium, Claviceps, Emericella, Neothyphodium*, and *Phoma* species is significantly higher (50–90%) than the relative amount of non-polar metabolites produced by “field” fungi from *Alternaria* and *Fusarium* species (25–50%) or “storage” fungi from *Aspergillus* and *Penicillium* species (30–45%).

Based on the discussed data, it can be assumed that endophytic fungi, as a rule, produce more lipophilic mycotoxins than phytopathogenic or saprophytic fungi. The degree of polarity of mycotoxins with high fungicidal activity, produced by endophytic fungi is also of great interest. This interest is largely supported by data on the inhibition of the development of *Aspergillus flavus* and *Fusarium verticillioides* in growing maize by the metabolites of the endophytic fungus *Acremonium zeae*. The authors showed that two antibiotics isolated from the culture of *Acremonium zeae*, pyrrocidines A and B, have pronounced antibacterial and antifungal activity against *Aspergillus flavus* and *Fusarium verticillioides* [85]. In this case, it is known and should be stressed that both pyrrocidines are fairly hydrophobic compounds with the same values of distribution coefficients (Log Pow = 5.5) [63].

These data are consistent with the results of the study on pollution of five species of grassy pasture plants (fescue, festulolium, timothy, perennial ryegrass, hedgehog grass), their mixture with clover and timothy-alfalfa mixture by micromycetes, including endophytic ones, and their 16 metabolites before the first and the second mowing of raw grass materials [23]. The work was carried out in the northwestern region of the Russian Federation.

Before the first mowing (June), the fungi of genera *Cladosporium*, *Alternaria* and *Phoma* were among the leaders in the number of colony-forming units per 1 g of raw material (CFU/g). Moreover, all three types of raw material were contaminated by fungi to a similar extent. Before the second mowing (August), the crop from which in Russia usually forms the basis of the “winter” dairy cattle diet, changes were noted in the leading group. Fungi of the genus *Acremonium* replaced the genus *Alternaria* in the “first three”, and the degree of contamination in terms of CFU/g changed significantly. In the herbal mixture, this indicator increased by 1.86 times, in the clover-herbal mixture by 15.9 times, mainly due to *Cladosporium, Acremonium* and *Phoma*, and in the mixture of timothy and alfalfa contamination decreased by 2.26 times. As the cumulative CFU/g index diminishes, the types of feed tested can be distributed in the following order: the herbal mixture with clover (CFU/g = 552533), the herbal mixture (CFU/g = 80360), and timothy–alfalfa mixture (CFU/g = 16667) [23]. It should be noted that among the metabolites of the “leading trio” of fungi, lipophilic toxins may account for about 70% (see Appendix A). Based on these data, it can be assumed that the use of alfalfa seems to be more preferable when used in herbal mixtures as compared with clover.

### 3.2. Other Non-Polar Cattle Feed Contaminants

As already mentioned, cattle feeds, in addition to mycotoxins, almost always contain also PAHs and POPs [37,45,47,48,49,50]. The highest degree of bioaccumulation among these groups of chemical compounds is characteristic of POPs [46]. The fact is, that POPs are evolutionary atypical substrates for the xenobiotic metabolism system of the cytochrome P-450 family of the vertebrate liver. Before the start of large-scale human industrial activity in the twentieth century, they did not occur in nature at all, and their metabolic rates in this system, unlike PAHs or mycotoxins are extremely low and inversely proportional to the content of chlorine or bromine atoms in their molecules [86,87,88]. This may be due to the fact that the negative induction effects of halogen atoms (fluorine, chlorine, bromine) in the molecule of an aromatic compound can reduce the density of the electron cloud between adjacent carbon atoms (bond order). Therefore, the oxidation rate of such “electron-depleted” bonds by cytochromes of the P-450 family decreases significantly with increasing degree of substitution of hydrogen atoms for chlorine or bromine in molecules of typical POPs [89]. A certain contribution to the difficulties of the metabolism of these compounds, as is commonly believed, is made by steric hindrances of substituents (chlorine and bromine) which are bulkier than the hydrogen atoms. As an example of such a “difficult” metabolism in vertebrates, we can give dieldrin. It is formed in the hepatic xenobiotic metabolism system by oxidation of one (least shielded) double bond C6-C7 in the aldrin molecule with a formation of the corresponding epoxide. In this case, the oxidation of the second double bond C2-C3, shielded by two chlorine atoms, usually does not occur. Therefore, POPs largely accumulate in adipose tissue and to a greater extent than PAHs and mycotoxins are transferred to animal products.

### 3.3. The Use of Adsorbents to Reduce the Toxic Load of Feeds

A study of the work and reviews on the use of feed adsorbents to protect animals from toxins from feed suggests that a large mass of adsorbents that are offered in the markets and are used in practice, are of two major types [59]. The first of them includes inexpensive natural minerals, in most cases, various clays built from silicates or aluminosilicates and their combinations, mined by the quarry method, and not requiring special expenses for their production. The second more expensive type of adsorbents for mycotoxins are yeast cell walls and combination products based on them, which are a processed by-product of the production of beer and strong alcoholic beverages. The yeasts itself as a waste from these industries, in particular, often without pretreatment, other than drying and grinding, is also used as feed additives to increase the level of amino acids, vitamins and trace elements in diets of farm animals. 

As an example of the successful application of adsorptive feed additives to reduce the toxic load of mycotoxins contained in feed, we can bring aflatoxin B1—a highly toxic, moderately (non-)polar mycotoxin (Log Pow = 1.6), produced by *Aspergillus* fungi. The beginning of a systematic study of the mycotoxins after the dramatic case of poisoning of a large herd of young turkeys with a lot of peanut meal containing aflatoxin B1 in significant quantities is associated with this toxin [14,15,16]. Aflatoxins, due to their high prevalence in feeds in areas with a warm climate, high toxicity and carcinogenicity, have been the most studied mycotoxins since the 1960s. For this reason, the use of many mycotoxin’s adsorbents has been aimed at removing mainly aflatoxin B1 from the gastrointestinal tract of farm animals. Indeed, numerous studies, the results of which are discussed in reviews [54,57,59], have demonstrated a fairly high efficiency of traditional aluminosilicate adsorbents or adsorbents from yeast cell walls to reduce the toxic effects of this mycotoxin and reduce the degree of transfer of its main metabolite, aflatoxin M1 to milk [32,90,91]. However, the effectiveness of these adsorbents with respect to more lipophilic mycotoxins, such as zearalenone (Log Pow = 3.6) or ochratoxin A (Log Pow = 4.7), was, as noted, significantly lower [54,56,57,58,59,60,61]. Zearalenone (ZEA) due to its lipophilic properties and a significant negative impact on the processes of reproduction of farm animals in many studies serves as a benchmark for assessing the effectiveness of the use of feed adsorbents. So in the study, in which in vitro the degree of binding of ZEA with 27 adsorbents from different manufacturers, at a toxin:adsorbent weight ratio 1:20,000, was evaluated, was shown that even with such a high adsorbent load on the toxin, only 7 of 27 adsorbents demonstrated a measurable binding of ZEA (more than 70%). Among them, activated carbon and additives containing humic acids, but not aluminosilicates or yeast cells walls were noted [56]. 

In the study of the effectiveness of adsorbents for binding ZEA in the gastrointestinal model [60], it was shown that the only adsorbent used that could bind ZEA with a measurable capacity was activated carbon, but only at a concentrations from 0.5% to 2%. Such concentrations of adsorbents in feed (sometimes up to 5%) are often used in studies assessing their effectiveness, but are rarely used in real animal husbandry for economic reasons. Manufacturers of adsorbents usually recommend their use in doses of 0.1% to 0.2% by weight of the feed.

In the study conducted in vivo it was noted that traditional aluminosilicate adsorbents are not able to protect the broiler population from the toxic effects of polychlorinated pesticides (typical POPs) [36]. The absence in the available scientific literature of reports on positive examples of the use of aluminosilicate adsorbents or adsorbents from yeast cell walls to relieve symptoms of the so-called “ryegrass staggers”, which are known to be triggered by lolitrems, lipophilic mycotoxins of endophytic *Epichloë* fungi [70], may indicate a lack of effectiveness of such adsorbents in this particular case. To combat this phenomenon, they usually follow the path of breeding new varieties of herbs with a low content of tremorgenic toxins. In most cases, such attempts fail. Another, less costly approach involves the use of non-polar feed adsorbents, which in this case seems more promising.

### 3.4. The Use of Non-Polar Adsorbents to Reduce the Toxic Load of Feed

In light of the above and based on the principle known in medicine “similia similibus curantur”, there is an urgent need to use adsorbents of another type, namely non-polar ones to protect animals from lipophilic toxins in feed. Such adsorbents are currently presented on the market mainly in two groups [59]: (1) activated carbons and (2) cholesteramine, based on porous polystyrene. Cholestyramine-based adsorbents are too expensive to be used in agriculture. Adsorbents based on activated carbon are not widely used because of their low capacity and, as a consequence, its inclusion in feeds in too high-doses. Often this is not economically viable. But, nevertheless, in studies evaluating the comparative capacity of sorbents in vitro, it was noted that the measurable efficiency towards ZEA (lipophilic toxin), as already mentioned, showed only activated carbon, and not aluminosilicate adsorbents or yeast cell walls [60], as well as some adsorbents containing humic acids [56]. When studying the ability of different adsorbents to bind in vitro highly toxic lipophilic mycotoxin ochratoxin A (Log Pow = 4.7) with a toxin:adsorbent ratio of 1:500, comparable to activated carbon and cholestyramine results showed “Myco AD A-Z”, as well as the combined adsorbent “Standard Q/FIS” consisting of a mixture of activated carbon, bentonite, yeast extract and aluminosilicates [61].

One more type of non-polar adsorbents is known, which were specially developed for the adsorption and separation of moderately lipophilic and lipophilic substances in high-performance liquid chromatography. We are talking about hydrophobized adsorbents on a polysilicate basis. Chromatographic adsorbents are usually obtained by treating specially synthesized, washed and dried porous silica gel particles (SiO_2_) of suitable size with the desired pore diameter in an anhydrous conditions with various reagents to obtain a covalent bond between the silica gel surface and the alkyl residues (-CnH2n+1), usually containing from 4 to 18 carbon atoms. These adsorbents, called reversed-phase (RP), are highly hydrophobic and are capable of effectively adsorbing in an aqueous medium any organic compounds with a partition coefficient greater than zero (Log Pow > 0). From the theory and practice of liquid chromatography, it follows that the binding strength of non-polar sorbates with such adsorbents is directly proportional to the value of the sorbate’s partition coefficient in the octanol/water system (Log Pow). RP-adsorbents are actually a solid-phase version of the partition of organic substances by lipophilicity in this system. First of all, this refers to adsorbents containing the octyl residue (-C_8_H_17_) as the alkyl substituent.

Aluminosilicate adsorbents and yeast cell walls have long been used in practical animal husbandry, but there is still no single theory that could explain the mechanisms for binding mycotoxins to their matrices. In the literature, many mechanisms of interaction of mycotoxins with these matrices are discussed, complex geometric models of the layered or spatial-crystalline structure of aluminosilicates and the geometric correspondence of the sizes of mycotoxin’s molecules to the distance between layers of phyllosilicates or the size of voids in the crystal lattice of tectosilicates are constructed. Almost all types of intermolecular interaction are also attracted to describe these mechanisms. All this is widely discussed in research articles and reviews on the topic [54,55,56,57,58,59,60,61]. However, the authors of these studies agree that a unified approach to explaining the preferred interaction of matrices with mycotoxins, and the rules by which one could predict the effectiveness of a specific adsorbent in relation to a particular toxin have not yet been developed. An overwhelming number of results with these adsorbents were obtained and are obtained empirically.

The model of hydrophobic interaction of a solute with a non-polar RP-matrix in the aquatic environment looks less complicated and relies on a single concept - minimizing the free energy of the system by maintaining the integrity of the water structure. The essence of the hydrophobic interaction is that lipophilic substances that are not capable of forming enough hydrogen bonds with water molecules violate its structure, and it is energetically more profitable for the system to bring such substances by means of Brownian motion either to the interface or to any hydrophobic surface inside the system and thus to restore the structure of water in the system. In this model, the binding strength of non-polar substances with the RP-matrix is barely related to the size and shape of the molecule and is directly proportional to the degree of substance lipophilicity, or Log Pow value [59,75,92,93]. For this reason, this concept has the ability to predict. In practice, this means complete prediction confidence, that ochratoxin A (Log Pow = 4.7), lolitrem B (Log Pow = 5.8), enniatin A1 (Log Pow = 7.4) or beauvericin C (Log Pow = 9.5) will be bound to the non-polar RP-matrix stronger and more efficiently eliminated from the alimentary tract than, for example, aflatoxin A1 (Log Pow = 1.6) or zearalenone (Log Pow = 3.6), and PAH benzo[a]pyrene (Log Pow = 6.0) or POPs p, p-DDT (Log Pow = 6.9) or dioxin (Log Pow = 6.4) will be bound stronger than PAH naphthalene (Log Pow = 3.3), or POP endrin (Log Pow = 3.7).

It should be noted, however, that RP-adsorbents on a polysilicate basis, that have successfully proven themselves in liquid chromatography, cannot be used as efficiently in agriculture as feed additives for at least two reasons. The first is the high price. The second reason lies in the relatively low capacity of such adsorbents in the aquatic environment. Activated carbon in an aquatic media also manifests similar properties. Unlike hydrated aluminosilicate adsorbents and yeast cell walls, RP-adsorbents in the form of dry matter (xerogel) and dry activated carbon are not able to swell in an aqueous solution. Due to the high hydrophobicity of the outer and inner surfaces, these substances are poorly or not at all moistened with water; and water, due to the high surface tension and small diameter of the internal pores, does not penetrate within the adsorbent particles. In chromatography, this problem is solved with the help of mobile phases containing water and organic solvents in different proportions, which better wet the surface of RP-adsorbent particles, and the wetting of internal pores is achieved by applying to the column with adsorbent an external pressure from 10 to 400 bar which allows to overcome surface tension forces and fill the internal pores of particles with a mobile phase. Therefore, at atmospheric pressure in an aquatic solutions, only the outer surface of RP-adsorbents and activated carbon, the value of which depends on the particle size, can have a measurable sorption capacity.

Recently there have been reports on the use in agriculture of a reversed-phase polysilicate adsorbent made according to another technology and produced as a partially hydrophobized polysilicate hydrogel containing hydrophobic octyl groups covalently bound to a hydrated insoluble polysilicate matrix [24,94]. Since this adsorbent is manufactured in an aqueous medium and it is a hydrogel, it does not need time for swelling, is initially well wetted by saliva, and begins to “work” in the mouth of animals and is thus able to more effectively protect the oral cavity, esophagus and rumen from the action of toxins.

This RP-adsorbent No. 4 (Figure 2) based on a polyoctylated polysilicate hydrogel (POPSH) in vivo has also demonstrated high efficiency in a herd of lactating cows [95]. For example, the use of this adsorbent in a herd of Holstein-Frisian lactating cows led to a significant decrease in SCC and to an increase in milk yield in comparison with the control group for 40 days. At the same time, the rate and degree of decrease in SCC (−64%) in milk and the increase in milk yield (+11%) were higher [95] than those observed when other “traditional” adsorbents were used under similar conditions [65,90,91,96,97]. Despite this fact, final conclusions on the degree of effectiveness of the use of one or another adsorbent can be made only after conducting joint comparative tests of adsorbents under the same conditions and on the same feed base.

As one more example of the successful use of non-polar adsorbents in vivo, data on the use of activated carbon and humic acids to control *Clostridium botulinum* in dairy cows can be cited [98].

It can be assumed that, at present, control over non-polar mycotoxins, PAHs and POPs is becoming particularly urgent, since they are only slightly absorbed by the “traditional” adsorbents of mycotoxins. At the same time, it is non-polar toxins that are capable of bioaccumulation, and their concentration in adipose tissue and their degree of influence on animal health, as well as transfer to milk and other animal products can significantly increase with long-term consumption of feed, even with low contamination levels. It is necessary to take into account the kinetic parameters of the enzymatic systems of the vertebrate liver detoxification system. It can be assumed that the lipophilic toxins that are present in the feed in low concentrations and their concentrations in the bloodstream are significantly lower than those necessary for the implementation of enzymatic transformations with a noticeable or optimal speed (Michaelis constant), and have every chance to go through the animal’s liver without biotransformation and reach unchanged fat depots for deposition. As a result of long-term use of food contaminated with non-polar toxins, sooner or later a situation arises when, due to bioaccumulation, the concentration of these toxins in adipose tissue and in the bloodstream is converted into toxic with all the negative consequences for the body. Therefore, the primary task for the chemical safety of feeding animals is to block the flow of lipophilic toxins from the gastrointestinal tract into the bloodstream, even at their low concentrations in feed. Hence, it is necessary, along with traditional adsorbents, to use non-polar adsorbents such as activated carbons, adsorbents containing humic acids or POPSH, as well as their combinations. There is also an urgent need for the development and implementation of new efficient and cost-effective non-polar feed adsorbents in the practice of dairy farming.

As already mentioned, this work demonstrated the high efficiency of this adsorbent for reducing the transfer to milk of lipophilic chlorinated pesticides aldrin (Log Pow = 4.5), dieldrin (Log Pow = 3.7) and heptachlor (Log Pow = 4.3), which are typical POPs (see Table 1). During 2018, the POPSH testing was carried out in several dairy farms in Moscow and Kaluga regions. Under tests the content in raw milk of chlorinated pesticides: isomers of hexachlorocyclohexane, DDT and its metabolites, DDD and DDE, as well as aldrin, dieldrin and heptachlor was also determined. In appreciable quantities, pesticides (DDT, DDD, DDE, aldrin, dieldrin and heptachlor) were found in the milk of the control group of a single farm located in the northeast of the Kaluga region. The concentrations of DDT and its metabolites were below the lower limits of quantification, so they were not included in Table 1. DDT and its metabolites, aldrin, dieldrin, and heptachlor in the experimental samples of milk from this farm after the application of POPSH for 40 days were not detected. This means that DDT (Log Pow = 6.9), DDE (Log Pow = 7.0) and DDD (Log Pow = 6.2) were also effectively removed from milk using POPSH. It can be assumed on this basis that its effectiveness in binding the most toxic tetra- and pentachlorobiphenyls, dibenzodioxins and dibenzofurans or polybrominated diphenyl ethers will not be lower than that of DDT and its metabolites, since the distribution coefficients of these substances range from 6.0 to 7.0.

By analogy, it is possible to predict with high probability that POPSH will also be able to effectively remove from the digestive tract of vertebrates the main causative factor for the development of the ryegrass staggers—lipophilic mycotoxin lolitrem B (Log Pow = 5.8). Confirmation will be obtained after a more complete study of this issue in practice.

It can be assumed that the use of POPSH or similar non-polar adsorbents will not only reduce the transfer of POPs into milk, but also reduce the contamination of other animal products (meat, eggs, caviar) with POPs and other lipophilic toxins, and further and help us to remove red meat from the WHO black list.

The use of non-polar adsorbents will presumably have a positive effect on the solution of problems associated with such negative phenomena in dairy farming as the “summer slump”, “summer mastitis”, “fescue toxicosis” or “ryegrass staggers”. Mycotoxins of endophytic fungi, which are most actively produced in the warm season, together with PAHs and POPs can play a significant role in their occurrence and development, along with heat stress. According to the data given in Appendix A, it is evident that among the metabolites of endophytic fungi, a significant part, as was mentioned above, is represented by non-polar substances, the share of which, among other metabolites, exceeds 50%. In addition, during the summer grazing period, dairy cattle are subject to an additional toxic load from PAHs and POPs, which are 100% non-polar compounds. In the warm season, with increasing temperature, sublimation into the atmosphere and transfer of dust particles containing PAHs and POPs to significant distances and their deposition onto the surface of grass and soil significantly increases [44,47]. While grazing, cows along with grass can also ingest some quantity of the soil particles (up to 1 kg per day). The content of PAHs and POPs in the soil can be hundreds of times greater than their content on the grass surface [47]. Probably for this reason, all the above-mentioned “summer disasters” are most pronounced at the end of the summer season, as a result of the gradual bioaccumulation of non-polar toxins during the warm period. We cannot also exclude from consideration the inhalation pathway of accumulation of PAHs and POPs in a pasture, when animals during grazing can inhale air from surface layers, which may at elevated temperatures contain high concentrations of sublimated PAHs and POPs. 

This is confirmed by literature data. So the authors of one of the article on the study of the effect of seasonality on the level of milk yields and SCC noted that the lowest level of SCC and the highest yields in the majority of the dairy farms in Florida are observed from February to April. They also called for new programs to improve the quality of milk, which should be focused on the conditions of animals between August and October, because at this time, most farms show a marked increase in SCC in collected milk and a decrease in milk yields [99].

### 3.5. Selection of the “Right” Adsorbents for the Protection of the Dairy Cattle Digestive Tract

Seeing as in practical dairy cattle breeding the standardization of feeds according to the level of toxic impurities is fundamentally impossible, it is advisable to use test panels of different adsorbents, both polar and non-polar, to determine the most suitable of them or their combinations, and the necessary dosages for each large enough feed lot. It should be noted that the dosage of the adsorbent is inversely related to the quality of the feed used, the concentration of toxic impurities and the period of toxic feed in use (bioaccumulation). Cattle present a convenient model for such testing, since even with quality care, they respond very flexibly to the quality of feed and clearly demonstrate this with the help of such an important indicator of milk quality as SCC.

It is known that SCC increases significantly with the use of toxic feed [65,90,91,95,96,97]. Most likely, this is due to the negative effect of toxic components of feed on the immune system of animals, whose function can be suppressed both by representatives of mycotoxins [66,82,83,84,100,101,102] and PAHs and POPs [103,104,105,106]. It is also known that many mycotoxins in vitro exhibit cytotoxic and cytostatic properties. A recent review on this topic compared the data on the biological activity of different mycotoxins in relation to model mammalian tumor cells, including human ones [107]. The information presented in this review allows us to conclude that dozens of mycotoxins exhibit in vitro antitumor activity in the micromolar concentration range, some for example, including austocystin D, brefeldin A, gliotoxin, leucinostatin A, ophiobolin A and wortmannin are already active in the nanomolar range, and at least one of them, for example, a metabolite of endophytic *Chaetomium* fungi, 11-epichaetomugilin I, shows cytotoxic activity against model tumor cells already in the picomolar range. In this regard, it can be assumed that the same cytotoxic activity of mycotoxins in vivo can be directed against rapidly dividing cells of the small intestine, as well as cellular elements of the immune system of animals, which can lead to digestive disorders and acquired immune deficiency and, as a result, to the development of inflammatory diseases. PAHs, POPs and their metabolites can also contribute to the development of immune deficiency in addition to, and in parallel with mycotoxins. Therefore, the need to use truly effective non-polar feed adsorbents is high.

In this model, the “ideal” adsorbent, which can be matched only empirically, is able to bind sufficient amounts of toxins, especially non-polar, in the gastrointestinal tract. Therefore, the high capacity of the adsorbent and the binding strength of the sorbate in an aqueous medium are very important. Depending on the age of the animals and the history of the use of contaminated feeds, in the first stage, it may be necessary to use higher doses of the non-polar adsorbents to remove lipophilic toxins from the fat depots. In the future, this dosage under the control of SCC can be adjusted downward. Effective from a practical point of view can be considered an adsorbent (or a combination of 2–3 adsorbents, one of which must be non-polar), which, when using available feed and applied dosages, is capable of maintaining SCC at the level of 80,000–120,000 cells/mL for a long time, and the use of which is economically proven.

It is known that with a decrease in SCC in bulk milk, an increase in milk yield is usually observed [1,2,11,95,97] that can serve as a bonus to justify the material costs of the purchase of adsorbents. An additional bonus in this case is also provided by a higher price for milk with lower SCC. In addition, with the effective protection of the animals from non-polar toxins with the help of the “right” adsorbents, we can expect an increase in the number of lactations in cows, and obtaining from them better-quality offspring and safer commodity products, namely meat and milk.

Since the choice of the “right” adsorbent and, especially, an effective combination of several adsorbents in a herd of lactating cows is a long, labor-intensive and rather expensive process, it seems appropriate to carry out a preliminary check of the effect of adsorbents of choice on feed toxicity in more simple biological systems. Such test systems may include toxicity tests on brine shrimp larvae [108,109], the protozoa [110] or on mammalian cell lines [107].

For the selection of the “right” adsorbents, we are primarily interested in the ratio of polar and non-polar toxins in feed samples. Therefore, the most suitable system for the extraction of toxins before determining the toxicity of feed seems to be the same system that is used to determine the degree of polarity of organic substances, the octanol/water system [74]. After extraction and centrifugation, this system is stratified, which simplifies the independent determination of the toxicity of polar and non-polar toxins in the aqueous and octanol phases, respectively. The parallel determination of the toxicity of these phases without the use of adsorbents and in the presence of adsorbents of different polarities in concentrations of 0.1–0.5% will make it possible to conduct a preliminary assessment of the effectiveness of different adsorbents or their combinations with minimal cost.

## 4. Conclusions

The ratio of polar, moderately polar and non-polar (lipophilic, or hydrophobic) mycotoxins was evaluated on a database of some physico-chemical properties of mycotoxins (*n* = 1500) formed in this work. It was shown that the share of lipophilic mycotoxins exceeds the share of polar (14%) and moderately polar (40%) mycotoxins, and is greater than 46%. Similar results were obtained when analyzing data from the literature on the contamination of real feeds for dairy cattle with lipophilic mycotoxins [20,26]. In addition, two more groups of lipophilic toxins—PAHs and POPs—are always present in dairy cattle feed [45,47]. It can be concluded that the presence of a greater variety of non-polar toxins in the feed of dairy cattle, as compared with feed for poultry and pig breeding indicates that non-polar feed adsorbents should be more widely used in dairy farms. This need is confirmed by the results of this work. It was shown that a new hydrophobic adsorbent based on POPSH is superior to “traditional” adsorbents, including activated carbon, in their ability to bind lipophilic toxins in vitro. We have previously reported that in vivo POPSH effectively protects lactating cows from non-polar toxins in their feed. This was reflected in a rapid decrease in SCC (−64%) and an increase in milk yields (+11%) [95]. In this study, it was established that in vivo in a herd of lactating cows, POPSH can significantly reduce the transfer of chlorinated pesticides, DDT and its major metabolites, and also aldrin, dieldrin and heptachlor into milk in a short time. This is highly likely due to their elimination from the gastrointestinal tract of animals with the help of the adsorbent. On this basis, it is highly likely that the use of POPSH may be useful for alleviating the symptoms of “ryegrass staggers” and other negative effects associated with the “summer slump”.

In this regard, it is necessary to pay special attention to the development and implementation of new and effective non-polar adsorbents in the practice of dairy farming. The results obtained allow to predict that after a thorough study of their properties and the wider use of non-polar adsorbents for protecting livestock against nonpolar toxins in feeds, an increase in terms of the productive life of dairy cows, the quality of repair heifers, as well the productivity, quality and safety of dairy and meat products for consumers can be expected.

All of the above may be useful in dairy farms that are not always able to maintain an adequate quality of food supply. It must be recognized, however, that these considerations have a practical meaning and the maximum benefit from the use of adsorbents can be obtained only in cases where all the regulatory conditions for the care and maintenance of animals are observed, and only the quality of the feed is critical.

## 5. Materials and Methods

### 5.1. Chemicals

Phosphate buffered saline (PBS, Cat.No.P3813, Merck KGaA, Darmstadt, Germany) and naphthalene (analytical standard, Cat.No.84679, 250 mg, Merck KGaA, Darmstadt, Germany) were purchased from Merck (Merck KGaA, Darmstadt, Germany). Zearalenone (Cat.No.3975/10, Tocris Bioscienses, Bristol, UK) was purchased from Tocris Bioscienses, UK and used without further purification. For the preparation of the mobile phase for HPLC, acetonitrile 2 grade (Cryochrome, LLC, Saint-Petersburg, Russian Federation, and distilled water were used. The aluminosilicate adsorbent and the adsorbent from yeast cell walls were purchased from local manufacturers’ dealers and did not undergo any treatment other than drying to constant weight at 105 °C. Activated carbon was purchased at the nearest pharmacy. Feed additive “Alvisorb”® (POPSH) was purchased from RPC “Fox and Co”, LLC (Moscow, Russian Federation).

### 5.2. Adsorbent Binding

The binding of sorbates with adsorbents was carried out in a solution of PBS with pH = 7.4, prepared in distilled water, at room temperature in a glass test tube. Naphthalene and zearalenone were dissolved in acetonitrile at a concentration of 1.0 mg/mL. The adsorbents (except of POPSH) were dried to constant weight at 105 °C and suspended with PBS at a concentration of 3.5 mg/mL on a magnetic stirrer. With constant stirring, from the suspensions of each adsorbent aliquots were taken and 6 g of the suspension was placed in three glass tubes with ground stoppers. In the control tubes 6 g of PBS was placed, and 0.1 mL of sorbate solution was added to each tube. The tubes were closed, the plugs were fixed using a Parafilm strip and the tubes were placed in a Multi Bio RS-24 rotary mixer (Bio San Ltd., Latvia). The samples were incubated for 3 h at room temperature and the turning speed of 30 rpm. Samples were centrifuged for 15 min at 2000 rpm and the concentrations of sorbates in the supernatant from all the tubes were analyzed by RP-HPLC. 

### 5.3. Chromatographic Analysis

The samples were analyzed in a chromatographic system consisting of a K-501 high-pressure pump, a JetStream Plus thermostat with an A1365 manual injector with a 20 μL loop, a K-2501 UV detector (Knauer GmbH, Germany) and the “Multichrom” data processing system (Ampersand, Ltd., Russian Federation). Column—Phenomenex Luna C8 (2), 3 μm, 4.6 × 100 mm (Part No.00D-4248-E0, Phenomenex Inc., Torrance, CA, USA). The analysis was carried out at a flow rate of 0.75 mL/min and a column thermostat temperature of 35 °C in an isocratic system acetonitrile:water = 60:40 (*v*/*v*). Naphthalene was detected at 221 nm and zearalenone at 236 nm. 

Quantitative determination was carried out according to the external standard method. Each tube was analyzed three times and the average value of the residual concentration of sorbate was calculated. The concentration of sorbates in the control tubes was taken as the initial concentration. All the results of measuring the concentrations of naphthalene and ZEA did not go beyond the linearity ranges of analytical methods used. The sorption capacity of the adsorbents for each of the sorbates was calculated from the difference between the initial and residual concentrations. To determine the sorption capacity of POPSH, its three exact weights were dried at 105 °C to a constant weight, and the capacity of all four adsorbents was calculated versus their dry weights.

### 5.4. Determination of Pesticides in Raw Milk

Cows from the experimental group (*n* = 65) received the feed additive POPSH together with feed during the morning feeding at a dose of 2 g per 1 kg of feed for 40 days. Cows from the control group (*n* = 65) received the same feed without additive. The conditions of housing, feeding, and milking of cows have been described previously [95]. On the day 40 of the experiment, samples of raw milk were taken in each group after morning milking, frozen and kept at a temperature not higher than −18 °C until analysis. Pesticides were analyzed by GLC-MS using standard methods. The limits of detection and quantification in the method used were 2 and 3 µg/kg, respectively. 

### 5.5. Tables of Partition Coefficients for Mycotoxins, Polyaromatic Hydrocarbons and Persistent Organic Pollutants

In the study to assess the degree of polarity of mycotoxins, PAHs, and POPs the calculated values of the partition coefficients in the octanol/water system (Log Pow): XLOGP3-AA (XLOGP3 pure atom-additive model) [62], presented in the PubChem database [63] for various chemical compounds, were used. 

The database on mycotoxins was initially based on the technical method for determining 243 mycotoxins using liquid chromatography with mass spectrometric detection [111], which cited the values of the calculated partition coefficients of mycotoxins and some other metabolites in the octanol/water system. Additional data on mycotoxins and producing fungi were obtained by analyzing scientific reviews and articles on the study of their properties [22,23,26,69,70,112,113,114,115,116,117] and other sources from the literature or Internet.

## Figures and Tables

**Figure 1 toxins-11-00256-f001:**
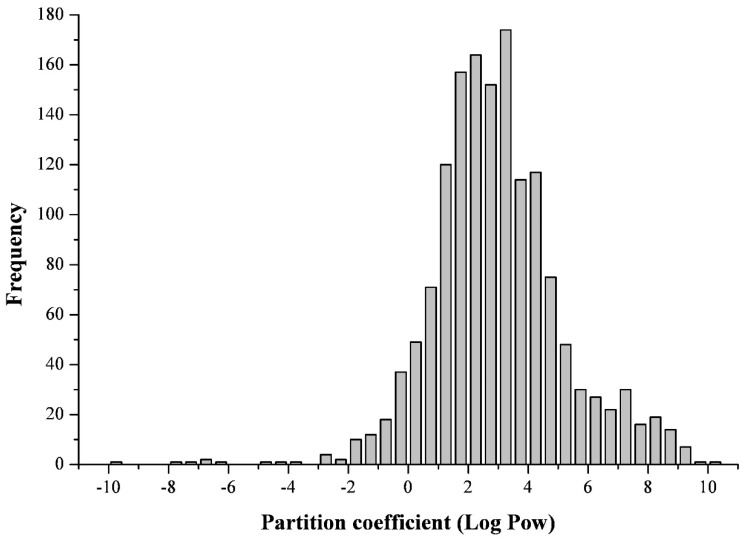
Distribution of some mycotoxins according to their polarity (lipophilicity).

**Figure 2 toxins-11-00256-f002:**
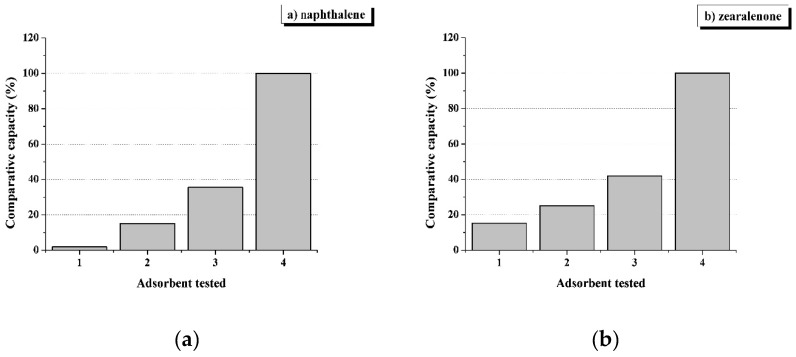
Comparative capacity of adsorbents of different nature in relation to lipophilic sorbates naphthalene (**a**) and zearalenone (**b**).

**Figure 3 toxins-11-00256-f003:**
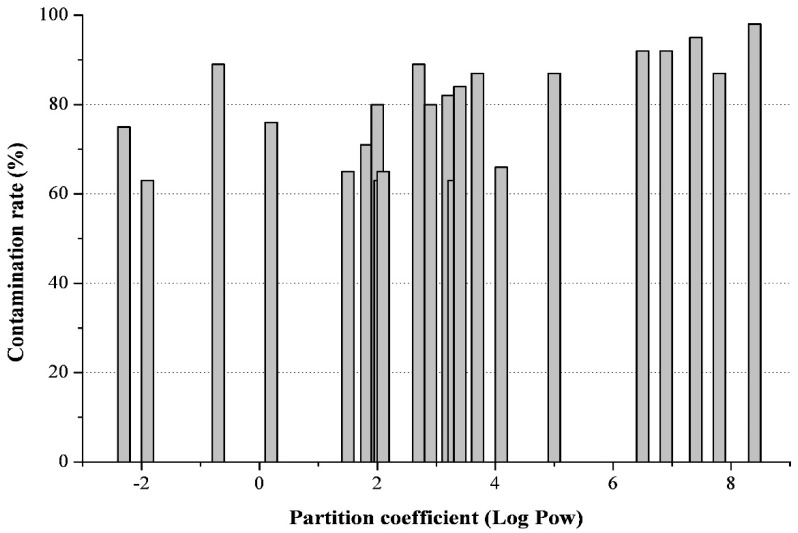
Contamination of dairy cattle feed with various mycotoxins (adapted from [26]).

**Table 1 toxins-11-00256-t001:** Concentration of chlorinated pesticides in raw cow milk.

Substance	Pesticide Concentration (µg/kg)	
	Control group	Experimental group
Aldrin	10.6 ± 0.35	n.d.*
Dieldrin	5.70 ± 0.21	n.d.
Heptachlor	5.85 ± 0.24	n.d.

* n.d. (not determined) means that the concentration of a substance in a sample does not exceed the limits of detection of the analytical method used (2 µg/kg).

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
