# Peer review of "Hydrophobized Reversed-Phase Adsorbent for Protection of Dairy Cattle against Lipophilic Toxins from Diet. Efficiensy In Vitro and In Vivo"

_toxins, 2019, doi:10.3390/toxins11050256_

Round 1
Reviewer 1 Report
This reviewers comments were addressed.
Author Response
Dear Reviewer 1,
First of all, we would like to thank you for the large, complex and tedious job in analyzing and assessing the quality of our article. I am very pleased that our manuscript did not cause any remarks. For me, this means that you belong to a rare class of high-level multilateral specialists and I feel that we definitely have common views on the level of approach to solving problems. So, I would like to contact you after our article will be published with a proposal for cooperation in research in this area, if it’s possible.
With a sincere sense of respect and gratitude,
A. S. (author for correspondence)
Reviewer 2 Report
The authors report on Diet.efficiensy of RP adsorbent for protection of dairy cattle against lipophilic toxins. They demonstrated in vitro and in vivo experiments using RP adsorbent. Although this manuscript is new approach of data science using internet and web, the sentences are redundant and it is difficult to understand what they want to claim. I do not recommend publishing the paper in its current form. My comments are:
1. Figures are not enough to persuade readers. Especially, more detail explain in figure legends. Table 1 needs each pesticide concentration in feeds using this experiment.
2. Deoxynivalenol and fumonisin are most contaminated mycotoxins in feed of worldwide. The author dose not recognize the danger of such mycotoxins (L297-300)
3. In Materials and methods, for the aluminosilicate and yeast wall, the author should be described their purification degree and origins.
4. There are many misspells ( ex. L420, L479) and some proper nouns are needed to explain ( ex.Mico AD AZ, L432).
Author Response
Dear Reviewer 2,
First of all, we would like to thank you for the large, complex and tedious job in analyzing and assessing the quality of our article. We fully agree with most of your comments and observations, and made the appropriate corrections to the text of the article, taking into account your recommendations. Provisions in which we can not fully agree with you, are in the table for the convenience of perception. In it, we allow ourselves to bring our arguments. For your convenience, we have arranged our answers opposite your comments on the items.
Remarks, questions, comments | Answers, responses, comments |
The authors report on Diet.efficiensy of RP adsorbent for protection of dairy cattle against lipophilic toxins. They demonstrated in vitro and in vivo experiments using RP adsorbent. Although this manuscript is new approach of data science using internet and web, the sentences are redundant and it is difficult to understand what they want to claim. I do not recommend publishing the paper in its current form. My comments are: | I agree. The main complaint that we put forward in our article is to draw the attention of the scientific community to a more detailed study of the role of non-polar toxins from feed in dairy farming. These toxins have so far been studied much worse than, say, aflatoxins, DON or fumonisins. On the other hand, non-polar mycotoxins from a database of 1,500 units (table S1), as shown, are more than 40%. And in the feed for dairy cattle, their share can be from 50 to 60% or more [20, 23, 26]. The lipophilicity of PAHs and POPs, which are present in almost all feeds for dairy cattle, can even not be discussed (Tables S2 and S3). References are given from the list of references of our article. |
1. Figures are not enough to persuade readers. Especially, more detail explain in figure legends. Table 1 needs each pesticide concentration in feeds using this experiment. | 1. We did not want to overload the article with excessive graphics, so we limited ourselves to those drawings that we considered sufficient for understanding the material. As for the content of the table 1, during the experiment we did not set ourselves the task of determining the carry-over rate of pesticides into milk during their model dosing. This is a topic for further research. The main goal of this experiment was to find out in principle whether the adsorbent POPSH is capable of removing pesticides from the body of cows. Therefore, we simply compared the content of pesticides in the milk of two groups of cows, which received the same diet, but one of them in addition received an adsorbent for 40 days. |
2. Deoxynivalenol and fumonisin are most contaminated mycotoxins in feed of worldwide. The author does not recognize the danger of such mycotoxins (L297-300) | 2. We do not diminish at all and are aware of the danger posed by polar mycotoxins like DON or fumonisins. But polar mycotoxins, as already mentioned (L297-300), can be excreted in the urine, unlike lipophilic toxins, which can gradually and imperceptibly to a certain concentration accumulate in adipose tissue. In addition, most polar mycotoxins can effectively, as stated by their manufacturers, be bound with "traditional" adsorbents such as aluminosilicates or yeast cell walls. Besides, it is considered that the total presence of DON and fumonisins is more characteristic for grain crops than for grass. Thus, in the paper cited in the article, the lipophilic mycotoxins beauvericin and equisetin were present in all the samples of grass and monocerin, zearalenone and aurofusariun were present in ≥90% of the samples, while the contamination of the grass samples with fumonisin B1 was 11%, with deoxynivalenol - < 10% [20]. Similar results were also obtained in the analysis of mycotoxin contamination of feed for dairy cattle and its components [26]. Fumonisin contamination of feed, as noted there, was about 20%. At the same time contamination of feed samples by eleven lipophilic mycotoxins ranged from 60 to 100% (nine of them – from 80 to 100%) as can be seen in Fig. 3. Therefore, we insist that the combination of lipophilic toxins (lipophilic mycotoxins, PAHs and POPs) performs a greater danger for dairy cattle than polar mycotoxins, which, of course, also make a certain contribution to the overall toxicity of feed. Regarding your concern about the role of DON and fumonisins in toxicoses, then in my opinion the best way to deal with them is to use a combination of several adsorbents, both polar and non-polar. |
3. In Materials and methods, for the aluminosilicate and yeast wall, the author should be described their purification degree and origins. | 3. I agree. Thanks to your commentary, we supplemented the Materials and Methods section with the phrase: The aluminosilicate adsorbent and the adsorbent from yeast cell walls were purchased from local manufacturers' dealers and did not undergo any treatment other than drying to constant weight at 105°C. (Marked in yellow). When conducting a comparative analysis of the adsorbents capacity, some researchers indicate brands of adsorbents and their manufacturers, for example, [55, 60,61], and some [56] - no. We chose the second way, because we didn’t want a conflict of interest in this regard. |
4. There are many misspells ( ex. L420, L479) and some proper nouns are needed to explain ( ex.Mico AD AZ, L432). | I agree. You are absolutely right. It is practically impossible for a researcher to avoid minor mistakes when writing an article. We have them all, thanks to your notes, corrected. All of them and other errors found in the second review in the text of the article are marked in yellow. |
I would very much like to hope that our explanations will satisfy you.
With a sincere sense of respect and gratitude,

Round 2
Reviewer 2 Report
In the first review, I suggested 4 comeents, but some comments was not understood my intention to the author.
1. Even though the author did not want to study carry-over of these hadards, it is necessary to make sure the present of these hadard in feed which was used in this experiment. The author should show that these hazards present in the feed.
3. I pointed out the purification degree, not brand name. The author should show the purification degree, example, the aluminosilicate contained >90 % in the absorbent.
Author Response
Dear Reviewer 2,
I apologize, but it seemed to me that I understood your motives and intentions regarding the article in question. Therefore, we express our sincere gratitude for your additional attempts to help us. As in our previous reply to your comments, I will again answer in the form of a table (for convenience).
1. Even though the author did not want to study carry-over of these hazards, it is necessary to make sure the present of these hazard in feed which was used in this experiment. The author should show that these hazards present in the feed. | Our plans really did not include a study of the carry-over rate of chlorinated pesticides into milk. This is a quite separate and difficult topic. I already wrote that in the future we may, in cooperation with other researchers, study this question on separate “model” pesticides, such as DDT, aldrin etc. Then the concentration of pesticide in the feed will be known and important for conducting a correct experiment. As for measuring the concentration of pesticides in feed, (at this stage) we did not consider this as necessary. For us, their presence in the milk of the control group of animals, which obtained no adsorbent, means that they are present or were present in the feed recently and have not been completely removed from the body with milk. And that's enough. We do not know how long the cows in this farm received contaminated feed, what was the level of contamination and how it historically changed. Such a hystory can not affect the results of this study by no means. We believe that at this stage of exploring the possibilities of using the non-polar adsorbent POPSH in dairy farming to reduce the concentration of pesticides in the milk is sufficient for planning further research in this direction. We also believe that for the time being, to preliminary evaluate the potential of a new adsorbent, a “yes or no” answer is quite enough. |
3. I pointed out the purification degree, not brand name. The author should show the purification degree, example, the aluminosilicate contained >90 % in the absorbent. | None of the manufacturers of feed adsorbents never indicates the degree of purification of one or another aluminosilicate material, which is produced as a feed additive. And in our study we used not some kind of aluminosilicate of reactive grade, but a real feed additive on an aluminosilicate base, which was purchased, as indicated in the "Materials and methods", from an authorized dealer of the manufacturer of this feed additive. Feed adsorbent is not a chemical reagent. We believe that there is no need to talk about the degree of purification of an aluminosilicate mined by an excavator in a quarry. Also, none of the articles on the study of the sorption properties of feed adsorbents indicating the brand or indicating only the type of adsorbent that are cited in our article does not indicate their composition and no hint on the degree of purification can be found. At best, the manufacturer indicates the approximate composition of its product. For the aluminosilicate adsorbent used in the study, the manufacturer indicates the following composition: bentonite-montmorillonite (50.6-60.6%), sepiolite (40.4-48.4%). All of the above applies to adsorbents based on yeast cell walls. |
I sincerely believe that our responses to your current remarks are quite enough so that, with your permission, we do not have to make additional changes to our article. I also hope that my answer will finally convince you of the modest merits of our article and you will not interfere with its publication. With sincere gratitude and respect,
A.S.
author for correspondence
This manuscript is a resubmission of an earlier submission. The following is a list of the peer review reports and author responses from that submission.
Round 1
Reviewer 1 Report
General comment
In my opinion this work is too wide and not enough detailed on some key points.
Occurrence should clearly be separated from toxic effects. Nearly nothing was said about mechanisms by which the studied compounds can have an effect on sub-clinical mastitis (having cumulative properties does not means these compounds have toxicity).
Nothing was said about metabolism of studied compounds whereas that point is a key point to understand cumulative properties and toxic effects, specially for mycotoxins.
Paragraphs on adsorbents is difficult to follow: several mechanisms are involved in the efficiency of adsorbents, not only non-specific binding linked to polarity.
See also below for detailed comments
Detailed comments
L5-21 : very difficult to understand the link between title and abstract. When I read title, I expect data on what could increase sub-clinical mastitis in non-polar toxins, i.e. toxic effect or dysregulation of cells involved in the defense against the micro-organisms involved in sub-clinical mastitis. When I read the abstract I believe that the manuscript will be on biological accumulation of toxic compounds with possible consequences on milk quality and safety for human consumers. In both cases, metabolism of toxic compound is a key point that needs specific investigation. This is true in all animal species but this has a special interest in cattle, especially in the understanding of toxic effects of mycotoxins.
L44-46: Indeed perhaps, so investigation on milk composition (finding these compound) should be done to strengthen this hypothesis: where is this bibliographic analysis?
L91-122: and about the metabolism of mycotoxins, both in rumen (degradability) and liver?
L128: what about the name of this compounds and their occurrence of mycotoxins? Cannot be as supplementary data as it is a key point in the risk to see an effect
L167-170: again, what about metabolism? Zearalenone is not the mycotoxin found in the plasma/tissues of cattle
L171-183: adsorbents is a big topic that justifies a bibliographic review alone. Different mechanisms of action of binder should be explained
L186-191: yes, studies on prevalence of 139 mycotoxins, so what about the 1361 (1500-139) other compounds?
Which of the 139 mycotoxins are non-polar? What is the metabolism of these non-polar compounds in cattle?
Which metabolites are found in milk? Which of them are known to have immunomodulatory effect?
L194-210: ok, so your analysis should firstly focus on compounds for which data in feed or raw materials are available. This should also include an analysis of the metabolism of these compounds in cattle. Then other mycotoxins can be compared to the most known compounds
L211-216: Very difficult to accept such paragraph. The risk to observe adverse effects is linked to the prevalence of a compound. The danger is linked to toxic properties. In cattle it is commonly accepted that the danger is reduced (in comparison to other vertebrates) because of degradations of mycotoxins by ruminal microflora. This topic is too big and complex to be summarized in 3 sentences. It needs long development as metabolized compounds have different properties from the parent compound, especially true and known for mycotoxins
L262: same general comment: occurrence and toxicity should be clearly separated. Chemicals biotransformation in cattle should be studied
L290: too wide, as said by authors there is several reviews on this topic. What can be studied here is the link between polarity and mechanism of action of adsorbents: L316-322.
L323: Activated carbon is not used not because of economic reasons but because of high level of incorporation is required, which makes the feed black, and because of wide nonspecific effect on other not toxic compounds, some of them having important nutritional properties.
L329: Difficult to accept data in broilers in a work on sub-clinical mastitis in cattle as oral bioavailability and metabolism are very different in the two groups of species
L336: difficult to understand the different paragraphs.
L418: as mycotoxins strongly varies in their properties it is probably very difficult to find one adsorbent able to bind all the mycotoxins. Authors should separate adsorbents depending on their mechanism of binding.
L420-424: What about metabolism? For most of the mycotoxins the parent compound is not found in milk, whereas traces of metabolites are detected.
Reviewer 2 Report
Specific comments to be addressed in revision:
The authors use the term "non-polar" to describe mycotoxins with a certain LogP value. There are several issues with this.
*****Log P is not a good indicator of non-polar properties for a weak acid like zearalenone. The "nonpolar" mycotoxin zearalenone described in this paper is hydrophobic under only specific conditions and it is slightly polar under those conditions, and very polar under other conditions. For example, zearalenone is polar because it has phenolic hydroxyls that can be deprotonated into a very polar form under mild conditions (pH 7+). Zearalenone possess a dipole moment 3.66 that makes it polar in its neutral form (below pH 7). The anionic form of zearalenone is very polar (above pH 8) with a dipole moment of 15+ debeye. It is not accurate to characterize zearalenone as "non-polar".
*****There are many, many mycotoxins that are non-polar that are not included in this study. The title is misleading. The title should use be specific that the work in on Zearalenone and not all mycotoxins.
*****
In light of these comments, it is suggested the paper be completing rewritten in a more accurate manner. The title of the paper should be changed as well. For example, the focus could be the role of predicted lipophilicity/lipophobicity or hydrophobicity/hydrophilicity in zearalenone efficacy. A discussion on the polarity of zearalenone and effects of pH on zearalenone's polarity should be included.